# Randomised controlled trial of a Calcium Channel or Angiotensin Converting Enzyme Inhibitor/Angiotensin Receptor Blocker Regime to Reduce Blood Pressure Variability following Ischaemic Stroke (CAARBS): a protocol for a feasibility study

Thompson G Robinson,[1] William J Davison,[2] Peter M Rothwell,[3] John F Potter[2]

¹Department of Cardiovascular Sciences, University of Leicester, Leicester, UK
²Department of Ageing and Stroke Medicine, University of East Anglia, Norwich, UK
³Nuffield Department of Neurosciences, University of Oxford, Oxford, UK

**Correspondence to**
Dr William J Davison;
willdavison@doctors.org.uk

## ABSTRACT

**Introduction** Raised blood pressure (BP) is common after stroke and is associated with a poor prognosis, yet trials of BP lowering in the immediate poststroke period have not demonstrated a benefit. One possible explanation for this may be that BP variability (BPV) rather than absolute levels predicts outcome, as BPV is increased after stroke and is associated with poor outcomes. Furthermore, there is evidence of distinct antihypertensive class effects on BPV despite similar BP-lowering effects. However, whether BPV in the immediate poststroke period is a therapeutic target has not been prospectively investigated. The objectives of this trial are to assess the feasibility and safety of recruiting patients following an acute ischaemic stroke or transient ischaemic attack (TIA) to an interventional randomised controlled trial comparing the effects of two different antihypertensive drug classes on BPV. Secondary exploratory objectives are to assess if different therapeutic strategies have diverse effects on levels of BPV and if this has an impact on outcomes.

**Methods** 150 adult patients with first-ever ischaemic stroke or TIA who require antihypertensive therapy for secondary prevention will be recruited within 7 days of the event from stroke services across three sites. After baseline assessments they will be randomly assigned to treatment with a calcium channel blocker or ACE inhibitor/angiotensin receptor blocker-based regimen and followed up for a period of three months.

**Ethics and dissemination** Ethical and regulatory approvals have been granted. Dissemination is planned via publication in peer-reviewed medical journals and presentation at relevant conferences.

**Trial registration number** ISRCTN10853487.

## Strengths and limitations of this study

► To our knowledge, this is the first prospective randomised trial designed to assess the treatment of blood pressure variability (BPV) following acute ischaemic stroke/transient ischaemic attack.
► The protocol incorporates multiple blood pressure measurement methods.
► The chosen therapeutic interventions are in line with standard clinical practice for secondary stroke prevention.
► The trial is open label which could bias the analysis of treatment effects on BPV and any impact on stroke outcomes, but these are secondary exploratory outcomes in this feasibility trial.

hospital admission[1 2]; SBP <130 mm Hg being the guideline target for secondary prevention following stroke.[3] Increased poststroke BP is associated with poor prognosis[4 5] and may result from raised intracranial pressure,[6] increased sympathetic nervous system activity,[7] abnormal baroreceptor sensitivity (BRS),[8] haematoma expansion,[9] cerebral oedema[10] and a white-coat response.[11] A spontaneous BP decrease usually occurs 4–10 days after ictus,[12] but substantial BP reductions can be associated with cerebral hypoperfusion as a consequence of poststroke dysautoregulation.[13] We have previously reported that both increased 24 hours and beat-to-beat BP levels following acute stroke are associated with a poor prognosis.[14–16] Subsequently, data from the International Stroke Trial have suggested a U-shaped relation between baseline SBP (within 48 hours of stroke) and short-term (14-day mortality) and long-term

## INTRODUCTION
### Background
Raised blood pressure (BP) is common after acute stroke with at least 75% of patients having a systolic BP (SBP) >130 mm Hg at

(6-month death and dependency) outcomes; the lowest risk of death and dependency being at SBP 150 mm Hg.[17] However, there is conflicting evidence regarding acute stroke hypertension treatment. Data from randomised controlled trials (RCT) suggest that BP can be safely reduced after the acute stroke period, however, there seems to be no indication that doing so is beneficial.[18–23] Indeed, the Scandinavian Candesartan Acute Stroke Trial reported that it may actually be harmful, with a non-significant increased risk of poor 6-month functional outcome.[23] Therefore, Cochrane meta-analysis and guidelines state that optimal BP management in the context of initial stroke management remains uncertain.[3 24–26]

An alternative explanation for the lack of evidence that lowering elevated BP levels in acute stroke is beneficial may relate to the additional effects of BP variability (BPV).[27] Current hypertension guidelines predominantly focus on mean, usually casual, BP measurements, dismissing BPV as random and merely an obstacle to the reliable estimation of usual BP. However, on ambulatory or home BP monitoring, which are recommended for the diagnosis and management of hypertension,[28] mean BP is found to vary substantially,[29] with the extent of this variation associated with visit-to-visit variability in clinic BP.[30] Indeed, there are many examples to support the potential importance of BPV for vascular risk.[30] First, the predictive value of estimated usual SBP and stroke risk falls with age,[31] yet stroke incidence rises with age and the relative benefit of antihypertensive therapy is maintained in the elderly.[32] Second, an increased early-morning surge in BP is predictive of stroke, but is poorly associated with mean BP.[33] Third, other causes of transient hypertension are recognised triggers of vascular events, including sympathetic overactivity and orthostatic hypertension.[34] Fourth, in the majority of studies, there is no threshold of baseline SBP below which vascular risk stops falling (though evidence for BP below 115/75 mm Hg is very limited),[31] with antihypertensive therapy reducing risk even at 'normal' baseline SBP.[35] Fifth, 'white-coat' hypertension, a common example of situational BPV, is associated with long-term target organ damage independent of mean BP.[36] Sixth, though hypertension is a recognised risk factor for vascular dementia, there is limited evidence of reduced dementia risk in trials of antihypertensive therapy. However, a trial of calcium channel blockers (CCB), which have the most consistent effect on reducing BPV,[37 38] has shown a substantial reduction in the incidence of dementia.[39] Furthermore, in patients with Alzheimer's dementia BPV is increased compared with matched controls, with increased BPV being independently predictive of progressive cognitive decline in this patient group.[40] Finally, specific group differences in stroke risk are not accounted for by mean BP alone, for example, in black individuals.[41]

### Rationale for the study

In a retrospective analysis of RCTs in a transient ischaemic attack (TIA) population, visit-to-visit intraindividual

BPV was a risk factor for stroke independent of the mean 'absolute' BP level, and perhaps of greater significance.[30] Additionally, within-visit systolic BPV, based on casual BP measurements, was correlated with visit-to-visit systolic BPV, but was a weak predictor of future vascular events.[30] Importantly, in a separate analysis it was demonstrated that BPV is reproducible and independent of confounding factors.[42] Increased BPV may also be an important predictor of short-term outcome following acute stroke. Robinson and colleagues have shown that beat-to-beat systolic BPV was greater in acute stroke compared with controls,[43] and that high mean arterial and diastolic beat-to-beat BPV was associated with a worse prognosis.[15] Furthermore, in a post hoc analysis of the Tinzaparin in Acute Ischaemic Stroke Trial, high systolic BPV from three to six casual BP readings, taken within 48 hours of symptom onset, was associated with an increase in death or early neurological deterioration at day 10.[44] Conversely, a retrospective analysis of nearly 1000 patients in the Continue or Stop Post-Stroke Antihypertensives Collaborative Study and Controlling Hypertension and Hypotension Immediately Post Stroke trial did not demonstrate a significant association between systolic BPV based on two sets of three casual BP readings within 48 hours of stroke onset and 2-week death and dependency.[45] Overall, a recent meta-analysis reported that increased systolic BPV, measured early from stroke onset, was associated with poor long-term functional outcome.[46] Furthermore, increased BPV may also relate to poststroke cognitive outcomes with evidence suggesting an association with signs of cerebrovascular small vessel disease on neuroimaging,[47] and deterioration in cognitive test scores.[48 49]

Clearly, there is further scope to explore the relationship between BPV and outcome following acute stroke, in particular whether it has implications for therapeutic management in the immediate poststroke period. Rothwell's group have explored the differential effects of BP-lowering therapies on BPV in a hypertensive population.[37 38] Though clinical benefits with reduction in risk of stroke and coronary events were seen for all classes of antihypertensive agent, class-specific effects existed; CCBs reduce stroke risk to a greater extent than expected from mean SBP reduction alone, and beta-blockers (BB) to a smaller extent. A detailed analysis of the Anglo-Scandinavian Cardiac Outcomes Trial-Blood Pressure Lowering Arm, comparing an amlodipine versus atenolol-based regime, and the Medical Research Council trial, comparing an atenolol versus diuretic-based regime, reported opposite effects of CCB and BB on systolic BPV. In addition, this differential effect accounted for the disparity in observed effects on stroke risk and observed effects on mean SBP.[38] This was confirmed in a systematic review and meta-analysis of 389 RCTs which also demonstrated that BPV is reduced by non-loop diuretic drugs, but increased by ACE inhibitors (ACEI) and angiotensin receptor blockers (ARB).[37] Again, the effects on systolic BPV were correlated with effects on stroke risk

independent of differences in mean SBP.[37] Prospective trials to investigate these apparent medication class effects on BPV would be valuable, especially comparing CCB and ACEI/ARB which are typically the first-line antihypertensive drug classes. If, as anticipated, CCBs reduce BPV whereas ACEI and ARBs increase it this could be relevant after acute stroke, where normal cardiovascular autonomic and cerebrovascular autoregulatory pathways are impaired. BRS is important in the short-term regulation of the cardiovascular system, including BP, and is known to be impaired following acute ischaemic stroke,[8] and associated with poor short-term and long-term prognosis.[50] In addition, it is well established that cerebral autoregulation (CA) is impaired, particularly following moderate to severe stroke.[13] As a consequence, cerebral perfusion is pressure dependent, and therefore hypertensive episodes related to increased BPV may contribute to reperfusion injuries, for example, postischaemic oedema and/or intracerebral haemorrhage. Conversely, hypotensive episodes associated with increased BPV in the presence of impaired CA may lead to secondary ischaemia, particularly in the absence of a good collateral circulation.

In conclusion, increased BPV is associated with a greater vascular risk, independent of mean BP and may predict poor outcomes after stroke. Furthermore, commonly used antihypertensive agents have different class effects on BPV which may in part explain the overall differential effects on stroke risk for similar absolute reductions in mean BP in a hypertensive population. Trials to investigate the potential therapeutic targeting of BPV and any potential benefit of doing so in acute stroke would be useful to address gaps in the current knowledge base.

## Study objectives

The primary objective of this study is to determine the feasibility of recruiting patients with acute stroke and TIA into an interventional randomised trial comparing the effect of different antihypertensive medication regimens on BPV.

Secondary feasibility objectives are:
► To determine the viability of measuring changes in BPV from baseline to 21 (±7) days and 90 (±14) days by treatment arm.
► To assess compliance rates with BPV measurement methods.
► To assess compliance rates with the investigational treatments.
► To identify serious adverse events (SAE) associated with the interventions, including recurrent stroke/TIA, other cardiovascular events, death and hospital readmission up to 3 months.

In addition to the feasibility objectives, exploratory outcomes that may be used in a future definitive RCT will be investigated. The proposed primary exploratory outcome will be modified Rankin Scale (mRS) score at day 90.

Exploratory secondary outcomes are:
► mRS at day 21.

► National Institutes of Health Stroke Scale (NIHSS) at day 21.
► Mean BP at day 21 and day 90.
► BPV at day 21 day and 90.
► Montreal Cognitive Assessment (MoCA) score at day 90.

## METHODS AND ANALYSIS
### Study overview
This study is a randomised, multicentre, open-label parallel group study to determine the feasibility of conducting such a trial in a National Health Service (NHS) setting to investigate class effects of antihypertensive medications on BPV in patients with acute ischaemic stroke or TIA. The aim is to evaluate barriers to recruitment, identify potential safety issues and demonstrate that it is possible to detect differences in BPV over the proposed study duration. We also hope to investigate the potential therapeutic benefit of targeting BPV after acute ischaemic stroke/TIA in terms of functional outcome in order to help estimate the necessary sample size for a future definitive trial. A summary of the study design is provided in figure 1. Recruitment commenced in January 2018 and is ongoing. The trial was prospectively registered: International Standard Randomised Controlled Trial Number 10853487.

### Patient and public involvement
The trial was conceived and designed without the involvement of patients or members of the public.

### Trial participants
All adult patients with clinically definite first-ever ischaemic stroke or TIA within 7 days of onset will be considered for the trial.

### Inclusion criteria
► Age >18 years.
► Patients with first-ever clinically definite TIA and ischaemic stroke (NIHSS <10).
► Within 7 days of symptom onset (this criterion was initially within 72 hours of symptom onset, but was altered with a substantial amendment to the protocol to try and improve recruitment).
► Casual BP >130/80 mm Hg on repeat measurements.
► Ability to comply with randomly assigned BP-lowering regime and BP measurements.
► Able to understand written and verbal English.
► Able to give informed consent.
► Willing to allow his or her general practitioner (GP) and consultant, if appropriate, to be notified of participation in the study.

### Exclusion criteria
The participant may not enter the trial if any of the following apply:
► Known definite contraindication to BP-lowering regime or therapeutic agents.

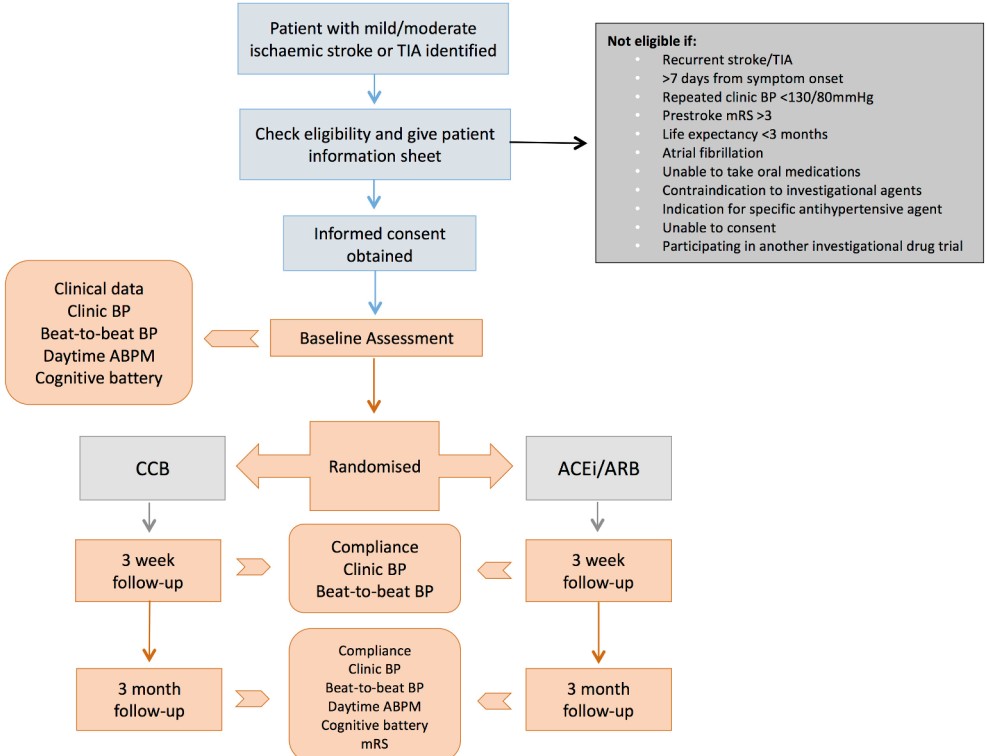

**Figure 1** Study flow diagram. ABPM, ambulatory BP monitoring; ACEi, ACE inhibitor; ARB, angiotensin receptor blocker; BP, blood pressure; CCB, calcium channel blocker; mRS, modified Rankin Scale; TIA, transient ischaemic attack.

▶ Swallowing difficulties which would preclude the taking of oral medication.

▶ Definite indication for BB, CCB, ACEI or ARB therapy.

▶ Significant prestroke dependency (mRS >3).

▶ Coexisting life-threatening condition with life expectancy <3 months.

▶ Previous participation in this trial or current participation in another investigational drug trial.

▶ Atrial fibrillation.

▶ Female participants who are pregnant, lactating or planning pregnancy during the course of the study.

▶ Unable to understand written and verbal English.

▶ Cannot give informed consent.

### Identification of participants

Patients with first-ever TIA and minor ischaemic stroke referred to and assessed by the inpatient and/or outpatient stroke services at three centres within 7 days of symptom onset will be identified by the treating clinician and/or the research team. If the patient provides verbal consent to be considered for the study then their medical records will then be assessed against the study inclusion and exclusion criteria. Patients known to be hypertensive and on treatment prior to their cerebrovascular event should have their antihypertensive medications suspended at admission, in keeping with standard practice at the recruiting centres, unless there is a specific indication for them to continue. Where treatment is suspended and the patient is willing to be considered for the trial then they are potentially eligible for inclusion provided other inclusion/exclusion criteria are not violated. Once

a potential participant has been confirmed to be eligible then research staff will approach the individual to discuss the study in more detail, provide a participant information sheet and seek written informed consent.

### Obtaining informed consent

The participant must personally sign and date the latest approved version of the informed consent form, countersigned by a delegated member of the research team, before any study-specific procedures are performed. Written and verbal versions of the participant information sheet and informed consent form will be presented to the participants detailing no less than: the exact nature of the study; the implications and constraints of the protocol; and the known side effects and any risks involved in taking part. It will be clearly stated that the participant is free to withdraw from the study at any time for any reason without prejudice to future care, and with no obligation to give the reason for withdrawal. The person who obtained the consent must be suitably qualified and experienced, and authorised to do so by the chief/principal investigator as detailed on the delegation of authority and signature log for the study. The original signed form will be retained at the study site within the investigator site file. A copy of the signed informed consent will be given to participants and a copy retained in their medical notes.

### Randomisation

After the baseline assessments eligible patients will be randomised using a computer-generated protocol, in blocks of 4, to a dihydropyridine CCB or ACEI/

ARB-based regime. The study treatment will be dispensed at the baseline visit, but treatment will not be commenced within 48 hours of the qualifying event in keeping with current recommended practice. The actual therapeutic agent used will be at the discretion of the treating clinician, but dictated by the class of therapy that the participant is assigned to. Prescription of the medication will be done by the treating clinician and the initial supply will be dispensed by the treating hospital or community pharmacy in accordance with the hospital's policy for providing discharge or outpatient medication. Further supplies will be provided by the participant's GP. Unblinding will not be necessary as there is an open-label study design.

## Interventions to be measured
### Routine clinical data
The following routine clinical information and investigation results will be obtained from the medical notes and by participant interview:
► Demographics (including age, sex, ethnicity, height and weight, smoking and alcohol habits).
► Medical history and family history of cardiovascular disease.
► Concomitant medications.
► NIHSS.
► mRS (including premorbid mRS).
► Oxford Community Stroke Project and Trial of ORG 10172 in Acute Stroke Treatment classification.
► Laboratory tests (including full blood count, clotting, urea, electrolytes, creatinine, estimated glomerular filtration rate, total cholesterol and random glucose).
► 12-lead ECG (±24 hours ECG if performed).
► Imaging investigations (including neuroimaging (CT or MRI), carotid ultrasound and cardiac echocardiography where applicable).

### BP measurements
Baseline casual BP will be calculated as a mean of two sets of three supine brachial BP readings taken 10 min apart using a UA767 BP monitor (referred to as enhanced casual BP).

Three consecutive periods of 10 min beat-to-beat non-invasive BP monitoring in the supine position using the middle finger of the non-hemiparetic hand will be recorded with a Finometer device. The servo adjust mechanism of the Finometer will be switched off during the recording period, but applied at 10 min intervals during the monitoring period.

Daytime ambulatory BP monitoring (ABPM) will be performed using a SpaceLabs 90207 monitor, programmed to measure BP at 20 min intervals. Daytime is defined as between 07:00 and 22:00 hours.

### Cognitive testing
A battery of cognitive tests will be performed. This will include the MoCA screening test which is established for use after cerebrovascular events, augmented with the Albert's line test for inattention, the Motor Neuron

**Table 1** Summary of trial procedures

| Procedures | Visits | | | |
|---|---|---|---|---|
| | Screening | Baseline | 21 (±7) days | 90 (±14) days |
| Informed consent | | X | | |
| Demographics | | X | | |
| Medical history | | X | | |
| Concomitant medications | | X | X | X |
| ECG | | X | | |
| Clinical investigation results (bloods tests, CT/MRI scan results) | | X | | |
| Eligibility assessment | X | | | |
| Randomisation | | X | | |
| Dispensing of study drugs | | X | | |
| Treatment compliance | | | X | X |
| Blood test for renal function in ACEI/ARB group | | X | | |
| NIHSS | | X | X | X |
| mRS | | X* | X | X |
| MoCA | | X | | X |
| Albert's line test | | X | | X |
| MiND-B | | X | | X |
| GDS | | X | | X |
| Enhanced casual BP | | X | X | X |
| Beat-to-beat BP measurements | | X | X | X |
| Daytime ABPM | | X | | X |
| SAEs | | | X | X† |

*Including premorbid mRS.
†SAEs at day 90 followed up until resolution.
ABPM, ambulatory BP monitoring; ACEI, ACE inhibitor; ARB, angiotensin receptor blocker; BP, blood pressure; GDS, Geriatric Depression Scale; MiND-B, Motor Neuron Disease Behavioural Instrument; MoCA, Montreal Cognitive Assessment; mRS, modified Rankin Scale; NIHSS, National Institutes of Health Stroke Scale; SAE, serious adverse event.

Disease Behavioural Instrument for frontal cognitive symptoms and the Geriatric Depression Scale to exclude significant concurrent anxiety/depression.

## Follow-up assessments
These will be undertaken at day 21 (±7 days) and day 90 (±14 days) in the trial centre or where the patient is resident at the time (including the hospital ward, rehabilitation facility or their own home). Interventions that will repeated at these follow-up visits are summarised in table 1. Additional follow-up interventions to assess the trial feasibility and safety will include assessment of treatment compliance using a self-reported questionnaire and tablet count (with compliance defined as ≥80%), and assessment of any side effects and SAEs. Patients randomised to the ACEI/ARB arm will have repeat renal function blood tests at the first follow-up visit in line with standard practice to ensure their safety. In

those patients failing to reach casual supine/sitting BP target of <130/80 mm Hg, the medical assessor at the follow-up visit will advise about altering BP-lowering treatment and this will be communicated to the participant's GP. The first-line change will be to increase the study regime medication (ie, CCB or ACEI/ARB) to twice the starting dose. If the patient is on the maximum dose of the study regime medication already, then the second-line change will be to add a thiazide-like diuretic. If a third-line change is required then spironolactone or an alpha-blocker will be added to the combination of study medication and thiazide-like diuretic. After the second follow-up visit ongoing management of the patient's BP will be taken over by the GP.

## Outcome measurements

### Primary feasibility outcome measure

Recruitment and retention rates at 3 months from the screening and management logs, and reasons for ineligibility or non-inclusion of those screened but not recruited.

### Secondary feasibility outcome measures

A. Changes in BPV from baseline to 21 (±7) days and 90 (±14) days by treatment arm.
B. Proportions of participants achieving ≥80% treatment compliance by treatment arm.
C. Treatment discontinuation rates.
D. Completion and failure rates of BPV measurements at 21 (±7) days and 90 (±14) days.
E. SAE rates by treatment arm.

### Exploratory outcome measures

A. mRS at 90 (±14) days by treatment arm.
B. mRS at 21 (±7) days by treatment arm.
C. NIHSS at 21 (±7) days by treatment arm.
D. Differences in mean BP at 21 (±7) days and 90 (±14) days by treatment arm.
E. Differences in BPV at 21 (±7) days and 90 (±14) days by treatment arm.
F. Differences in MoCA score at 90 (±14) days by treatment arm.

## Sample size calculation

A feasibility study of 150 patients (64 patients per group with a 15% dropout rate) will have an 80% power at the 5% significance level of detecting an 8 mm Hg difference in systolic BPV between the CCB and ACEI/ARB-based regimes, assuming a mean systolic BPV SD of 14.97 mm Hg in the CCB arm and 16.95 mm Hg in the ACEI/ARB arm.[37]

## Data analysis plan

The primary objective is assessment of feasibility. This will focus on recruitment and retention rates, compliance, change in BPV and safety of the intervention. Exploratory analysis of the effect of the proposed intervention on BPV and stroke outcome will be done as a secondary objective.

## Recruitment and retention

The total numbers of patients screened, the proportion recruited and the proportion completing follow-up will be determined. Reasons for ineligibility, non-inclusion and withdrawal will be analysed using descriptive statistical methods.

## Assessment of the intervention

Compliance with the intervention will be assessed by the proportion of participants who achieve ≥80% adherence to the trial medication and the proportion of participants who have all BP measurements recorded successfully.

The feasibility of detecting changes in BPV will also be assessed. Within-individual systolic, diastolic and mean BPV will be expressed as the SD, coefficient of variation, average real variability and variation independent of the mean calculated from all BP measurements: enhanced casual, beat-to-beat measurements (each 10 min recording and the total 30 min recording), and daytime ABPM.[42] Changes in within-individual BPV from baseline to the follow-up time points will be analysed using a general linear model. The size of the mean difference will be estimated for each approach and compared to select the most appropriate measure for a future study.

## Safety

Rates of SAEs, including recurrent stroke/TIA, other cardiovascular events, death and hospitalisation, will be recorded up to 3 months. A descriptive comparison will be undertaken to compare the rates, but no formal hypothesis testing will be undertaken.

## Exploratory analyses

Mean BP will be calculated from enhanced casual measurements. Change in mean BP from baseline to follow-up by treatment arm will be compared using an independent samples t-test.

An assessment of treatment effect on BPV will be undertaken stratified according to treatment arm. A general linear model will be used with BPV as the dependent variable and treatment arm as the independent variable, adjusting for baseline BP and diagnosis (stroke vs TIA). Each expression of BPV as described above will be analysed.

Exploratory assessment of treatment effect on stroke outcome will be undertaken by comparing between-group differences in mRS and MoCA score at follow-up using independent samples t-tests or a non-parametric test if the assumptions of the t-test are violated.

## Ethics and dissemination

This study was granted ethical approval in England (London–Central Research Ethics Committee, REC 17/LO/1427) and clinical trial authorisation from the Medicines and Healthcare products Regulatory Agency (EudraCT number 2017-002560-41). Subsequently, the trial was approved by the Health Research Authority. Study oversight will be conducted through regular meetings of a Trial Steering Committee and a separate Safety Committee, both of which will include independent representatives. If it is felt that the risk to participants

is significant or unacceptable the Safety Committee can recommend early termination of the trial.

The proposed investigational medicinal products are antihypertensives that are already in routine use and so their safety profiles are known. The medications are expected to lower the BP of participants. Therefore, in line with accepted stroke guidelines we will only recruit patients with uncontrolled BP (>130/80 mm Hg) who would otherwise require antihypertensive treatment for secondary stroke prevention. Medications that inhibit the renin-angiotensin system are known to potentially cause kidney dysfunction in patients with unrecognised renal artery stenosis. To ensure the safety of patients commenced on these medications a blood test for kidney function will be done at the 2–4 weeks follow-up which is in keeping with standard practice.

The trial will be conducted in accordance with the Declaration of Helsinki and the ICH Guidelines for Good Clinical Practice. All participants will provide written informed consent. Data will be collected and handled in line with sponsor standard operating procedures and NHS Trust policies. Electronic data will be anonymised and all data will be kept under secure conditions. TGR will act as data custodian.

Dissemination of the study results is planned via publication in peer-reviewed medical journals and presentation at relevant scientific conferences. Any reporting will adhere to the Consolidated Standards of Reporting Trials statement extension for pilot and feasibility trials. We do not intend to employ professional writers.

**Contributors** TGR, JFP and PMR obtained funding for the project and designed the study as part of a programme of work on blood pressure variability and stroke. The protocol was written by TGR and WJD and reviewed by the other contributing authors. This manuscript has been prepared by WJD and adapted from protocol version 2.0 dated 20 July 2018. The final manuscript has been reviewed by all authors and approved for submission/publication.

**Funding** This work was supported by a programme grant awarded jointly by The Stroke Association and The British Heart Foundation (Ref: TSA BHF 2012/01). The study is sponsored by the University of Leicester. Contact details for the sponsor: Dr Michelle Muessel (Research Governance Manager, Research Governance Office, Research and Enterprise Division, University of Leicester, Leicester General Hospital, Gwendolen Rd, Leicester LE5 4PW, Tel: 0116 2584099/2584867, email: uolsponsor@le.ac.uk).

**Competing interests** TGR and PMR are both NIHR senior investigators.

**Patient consent** Not required.

**Ethics approval** London–Central Research Ethics Committee (REC 17/LO/1427).

**Provenance and peer review** Not commissioned; externally peer reviewed.

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
