## [Reviewer comments · BMJ Open]

This paper was submitted to a another journal from BMJ but declined for publication following peer review. The authors addressed the reviewers' comments and submitted the revised paper to BMJ Open. The paper was subsequently accepted for publication at BMJ Open.

(This paper received three reviews from its previous journal but only two reviewers agreed to published their review.)

ARTICLE DETAILS

TITLE (PROVISIONAL)	A Randomised Controlled Trial of a Calcium Channel or Angiotensin Converting Enzyme Inhibitor/Angiotensin Receptor Blocker Regime to Reduce Blood Pressure Variability following Ischaemic Stroke (CAARBS) – a protocol for a feasibility study
AUTHORS	Robinson, Thompson G; Davison, William; Rothwell, Peter; Potter, John F;

VERSION 1 – REVIEW

REVIEWER	Simona Lattanzi Marche Polytechnic University, Ancona, Italy
REVIEW RETURNED	30-Jul-2018

GENERAL COMMENTS	The Authors described the protocol of a study aimed to assess the feasibility and safety of recruiting and randomizing patients following an acute ischaemic stroke or transient ischaemic attack to one of two antihypertensive regimens, namely a calcium channel blocker versus an angiotensin converting enzyme inhibitor/angiotensin receptor blocker, having different impact on BPV. Secondary endpoints will be the effects of the different therapeutic regimens on BPV and clinical outcome. This is the first prospective randomized trial designed to explore the treatment of BPV after an acute cerebral ischemia, and its findings are expected to be highly relevant from the clinical point of view. Some issue could be however further addressed. As Authors correctly stated in the background, hypertension is a recognized risk factor for dementia, but there is limited evidence of reduced dementia risk in trials of antihypertensive therapy. In this respect, the role of high BPV as a reliable risk factor for cognitive deterioration and progression of cognitive decline has been demonstrated in a number of observational studies (see: Visit-to-visit variability in blood pressure and Alzheimer's disease. J Clin Hypertens (Greenwich). 2018; Blood pressure variability in Alzheimer's disease and frontotemporal dementia: the effect on the rate of cognitive decline. J Alzheimers Dis. 2015; Visit-to-visit blood pressure variability in the elderly: associations with cognitive impairment and carotid artery remodeling. Atherosclerosis. 2014). This point may be briefly referenced in the background section to further highlight the notion that BP fluctuations, and not only
--

	absolute BP fluctuations, can have detrimental effects and represent a neglected therapeutic target. The impact of BPV following an acute episode of cerebral ischemia may vary according to a series of both structural and functional characteristics of any individual patient. Indeed, BP fluctuations could work synergistically with the preexistent burden of cerebral small angiopathy, leukoaraiosis and microbleeds, the status of extra-and intra-cranial vessel and cerebrovascular reactivity, and the baseline history of arterial hypertension, which can affect BP auto-regulation capacity and thresholds (see also Blood pressure variability and stroke outcome in patients with internal carotid artery occlusion. J Neurol Sci. 2014; Ischemic lesions, blood pressure dysregulation, and poor outcomes in intracerebral hemorrhage. Neurology 2017). Therefore, it would be useful to plan subgroup analyses, which could take into account how these variables may influence the effects of treatment on BPV and, eventually, stroke outcome.
--	--

REVIEWER	Mads Rasmussen, MD, PhD Associate Professor, Department of Anesthesia & Intensive Care, Section of Neuroanesthesia, Aarhus University Hospital, Denmark
REVIEW RETURNED	06-Aug-2018

GENERAL COMMENTS	Thank you for the opportunity to review the manuscript by Robinson TG et al. The authors performed a feasibility and safety study comparing the effects of a calcium channel blocker and ACE inhibitor/ARB regime on blood pressure variability in patients with TIA and minor ischemic stroke. Overall the manuscript is very well written. The background, rationale, objectives and methods including sample size calculation and statistical analysis are appropriate and clearly presented. Not being an expert in this particular field, it is unclear to me why the authors decided to compare the CCB regime with ACE/ARB's. From their literature review it appears that there are some data on the effects of CCB on BPV but not on the effects of ACE/ARB on BPV. In the "rationale for the study" section it is only mentioned that ACE inhibitors appear to increase BPV. It could improve this section if the supposed/expected differences in effects on BPV between the two drug classes were more clearly presented.
---

VERSION 1 – AUTHOR RESPONSE

Reviewer: 1

Reviewer Name: Simona Lattanzi

Institution and Country: Marche Polytechnic University, Ancona, Italy

Please state any competing interests or state 'None declared': None declared

Please leave your comments for the authors below

The Authors described the protocol of a study aimed to assess the feasibility and safety of recruiting and randomizing patients following an acute ischaemic stroke or transient ischaemic attack to one of two antihypertensive regimens, namely a calcium channel blocker versus an angiotensin converting enzyme inhibitor/angiotensin receptor blocker, having different impact on BPV. Secondary endpoints will be the effects of the different therapeutic regimens on BPV and clinical outcome.

This is the first prospective randomized trial designed to explore the treatment of BPV after an acute cerebral ischemia, and its findings are expected to be highly relevant from the clinical point of view. Some issue could be however further addressed.

Thank you for the positive comments.

As Authors correctly stated in the background, hypertension is a recognized risk factor for dementia, but there is limited evidence of reduced dementia risk in trials of antihypertensive therapy. In this respect, the role of high BPV as a reliable risk factor for cognitive deterioration and progression of cognitive decline has been demonstrated in a number of observational studies (see: Visit-to-visit variability in blood pressure and Alzheimer's disease. *J Clin Hypertens* (Greenwich). 2018; Blood pressure variability in Alzheimer's disease and frontotemporal dementia: the effect on the rate of cognitive decline. *J Alzheimers Dis*. 2015; Visit-to-visit blood pressure variability in the elderly: associations with cognitive impairment and carotid artery remodeling. *Atherosclerosis*. 2014). This point may be briefly referenced in the background section to further highlight the notion that BP fluctuations, and not only absolute BP fluctuations, can have detrimental effects and represent a neglected therapeutic target.

Thank you for highlighting these interesting and useful references. We have made the suggested addition to the background section (P.5).

The impact of BPV following an acute episode of cerebral ischemia may vary according to a series of both structural and functional characteristics of any individual patient. Indeed, BP fluctuations could work synergistically with the preexistent burden of cerebral small angiopathy, leukoaraiosis and microbleeds, the status of extra-and intra-cranial vessel and cerebrovascular reactivity, and the baseline history of arterial hypertension, which can affect BP auto-regulation capacity and thresholds (see also Blood pressure variability and stroke outcome in patients with internal carotid artery occlusion. *J Neurol Sci*. 2014; Ischemic lesions, blood pressure dysregulation, and poor outcomes in intracerebral hemorrhage. *Neurology* 2017). Therefore, it would be useful to plan subgroup analyses, which could take into account how these variables may influence the effects of treatment on BPV and, eventually, stroke outcome.

This is an interesting and pertinent point. As the mechanisms underlying increased BPV are yet to be fully elucidated there may well be additional patient characteristics that influence the response of variability to treatment that could be investigated. Given that this is a feasibility study and will likely have a relatively modest sample size, further breakdown of the data for subgroup analyses will probably result in samples too small to draw meaningful conclusions. However, if the feasibility trial indicates that we are able to proceed to a larger trial then we agree that planning for subgroup analyses (for example, patients with evidence of small vessel disease on neuroimaging, or those with chronic hypertension) would be valuable.

Reviewer: 2

Reviewer Name: Mads Rasmussen, MD, PhD

Institution and Country: Associate Professor, Department of Anesthesia & Intensive Care, Section of Neuroanesthesia, Aarhus University Hospital, Denmark

Please state any competing interests or state 'None declared': No competing interests

Please leave your comments for the authors below

Thank you for the opportunity to review the manuscript by Robinson TG et al. The authors performed a feasibility and safety study comparing the effects of a calcium channel blocker and ACE inhibitor/ARB regime on blood pressure variability in patients with TIA and minor ischemic stroke.

Overall the manuscript is very well written. The background, rationale, objectives and methods including sample size calculation and statistical analysis are appropriate and clearly presented. Not being an expert in this particular field, it is unclear to me why the authors decided to compare the CCB regime with ACE/ARB's. From their literature review it appears that there are some data on the effects of CCB on BPV but not on the effects of ACE/ARB on BPV. In the "rationale for the study " section it is only mentioned that ACE inhibitors appears to increase BPV. It could improve this section if the supposed/expected differences in effects on BPV between the two drug classes were more clearly presented.

Thank you for your positive comments. On reflection we agree that our choice of treatment arms in the trial could be explained more clearly. We have adjusted the second paragraph of the "rationale for the study" (P.6) to try and better explain our choice and explicitly state the anticipated difference in relation to BPV from the two treatment arms.

VERSION 2 – REVIEW

REVIEWER	Simona Lattanzi Marche Polytechnic University, Italy
REVIEW RETURNED	26-Sep-2018

GENERAL COMMENTS	The Authors have addressed the queries.
---

REVIEWER	Mads Rasmussen Department of Anesthesia and Intensive Care, Section of Neuroanesthesia, Aarhus University Hospital, 8000 Aarhus C, Denmark
REVIEW RETURNED	09-Oct-2018

GENERAL COMMENTS	The authors have addressed my concerns sufficiently and i recommend publication
---